# Sunken Riches: Ascomycete Diversity in the Western Mediterranean Coast through Direct Plating and Flocculation, and Description of Four New Taxa

**DOI:** 10.3390/jof10040281

**Published:** 2024-04-11

**Authors:** Daniel Guerra-Mateo, José F. Cano-Lira, Ana Fernández-Bravo, Josepa Gené

**Affiliations:** Unitat de Micologia i Microbiologia Ambiental, Facultat de Medicina i Ciències de la Salut and Institut Universitari de Recerca en Sostenibilitat, Canvi Climàtic i Transició Energètica (IU-RESCAT), Universitat Rovira i Virgili, 43201 Reus, Spain; daniel.guerra@urv.cat (D.G.-M.); jose.cano@urv.cat (J.F.C.-L.); ana.fernandez@urv.cat (A.F.-B.)

**Keywords:** *Ascomycota*, culture media, isolation, marine fungi, multi-locus phylogeny, taxonomy

## Abstract

The Mediterranean Sea stands out as a hotspot of biodiversity, whose fungal composition remains underexplored. Marine sediments represent the most diverse substrate; however, the challenge of recovering fungi in culture hinders the precise identification of this diversity. Concentration techniques like skimmed milk flocculation (SMF) could represent a suitable solution. Here, we compare the effectiveness in recovering filamentous ascomycetes of direct plating and SMF in combination with three culture media and two incubation temperatures, and we describe the fungal diversity detected in marine sediments. Sediments were collected at different depths on two beaches (Miracle and Arrabassada) on the Spanish western Mediterranean coast between 2021 and 2022. We recovered 362 strains, and after a morphological selection, 188 were identified primarily with the LSU and ITS barcodes, representing 54 genera and 94 species. *Aspergillus*, *Penicillium*, and *Scedosporium* were the most common genera, with different percentages of abundance between both beaches. Arrabassada Beach was more heterogeneous, with 42 genera representing 60 species (Miracle Beach, 28 genera and 54 species). Although most species were recovered with direct plating (70 species), 20 species were exclusively obtained using SMF as a sample pre-treatment, improving our ability to detect fungi in culture. In addition, we propose three new species in the genera *Exophiala*, *Nigrocephalum*, and *Queenslandipenidiella*, and a fourth representing the novel genus *Schizochlamydosporiella*. We concluded that SMF is a useful technique that, in combination with direct plating, including different culture media and incubation temperatures, improves the chance of recovering marine fungal communities in culture-dependent studies.

## 1. Introduction

Fungi constitute a significant portion of the global genetic diversity, estimated to encompass around 2.5 million species [1]. Currently, only around 155,000 species have been formally described, suggesting that over 90% of species remain undiscovered by science [2]. The marine environment is no exception to the dearth of knowledge on fungal diversity. This environment was considered poor for fungi throughout history based on the traditional approach to detecting fungal species, i.e., culturing on prepared media or incubated samples coupled with identification based exclusively on microscopy. However, after great sampling effort, the establishment of fungal barcodes for identification, and the development of culture-independent approaches like metabarcoding, the perception has shifted, and the concealed diversity of marine fungi is gradually being unraveled [3]. In fact, although the number of marine fungi is low in comparison to terrestrial ecosystems, the count of species identified in marine habitats continues to rise, with over 1900 species currently documented worldwide [4,5]. The number of marine fungal species and their respective descriptions can be accessed in repositories such as marine fungi (https://www.marinefungi.org/ (accessed on 20 January 2024); Ref. [6]).

In this context, the Mediterranean Sea stands out as a biodiversity hotspot [7]. Biodiversity hotspots are endangered habitats that have become international priorities for conservation efforts due to their great biodiversity and their role as repositories of undiscovered taxa [8]. In the Mediterranean Sea, various substrates have been studied in search of fungi, including water and sediment [9], driftwood and seagrasses [10,11], and even invertebrate animals [12]. Sediments have been the most studied substrate, revealing that the fungal community is dominated by the Phylum *Ascomycota* [9,11,13]. The unique geographic location and relatively higher temperature of the Mediterranean Sea compared to the oceans suggest that it may host a distinctive fungal community [14].

The prevalence of *Ascomycota* in Mediterranean sediments has been confirmed by metabarcoding, although only a small fraction of the detected ascomycetes could be confidently assigned to known species [15]. The effectiveness of fungal identification using this powerful technique ultimately relies on the quality and availability of barcodes in public databases [16]. Unfortunately, merely half of the known marine fungi are represented through DNA barcodes of either terrestrial or marine origin in GenBank [14]. Fungal DNA barcodes are derived from pure cultures, emphasizing the need for ongoing efforts in marine mycology to improve the tools for fungal detection in culture [17].

While yeasts are a well-documented group of marine fungi, owing to their ease of cultivation, knowledge regarding filamentous fungi is limited [14]. The challenges in culturing filamentous marine fungi stem from either the fungus’s incompatibility with synthetic media due to its lifestyle (i.e., endophytes, parasitic) or their low abundance in the environment. Given that the marine habitat is composed primarily of seawater, which can disperse fungal propagules, a potential solution to enhance the chance of culturing marine fungi could be to concentrate the environmental sample.

Skimmed milk flocculation (SMF) is a concentration technique designed to collect the microbial content within an environmental sample in flocs of skimmed milk. This technique was initially developed to improve the detection of viruses in seawater [18,19,20] and has subsequently been demonstrated to be successful in the detection of bacteria and protozoa [21]. In addition, SMF is recognized as a repeatable, cost-effective technique with broad applicability, such as monitoring water quality [21,22,23]. However, despite its proven efficacy in other microbial groups, the application of this technique for the detection of fungi remains unexplored.

In this work, we aimed to detect in culture as much diversity as possible of filamentous ascomycetes from marine sediments of the western Mediterranean Sea basin (Spain, Catalonia, Tarragona province). To accomplish this, we employed a dual approach involving direct plating of the sediment and a pre-treatment using SMF on the sample. We describe the culturable community of filamentous ascomycetes and assess the impact of each approach on detecting fungal diversity. This evaluation is complemented by the use of different culture media and isolation temperatures. In addition, we delineated the taxonomy of several interesting specimens under a consolidated species concept, resulting in the description of a new genus and four new species that belong to different classes of *Ascomycota*.

## 2. Materials and Methods

### 2.1. Study Area and Sampling

The Tarragona province is located on the western side of the Mediterranean Sea in the southern part of Catalonia, Spain. This area is known as the “golden coast” for its natural sandy beaches, which are a key attraction for tourism [24]. Notably, the port of Tarragona serves as a docking point for tourist cruise ships, ranking as the fifth most important harbor in Spain [25]. Home to a diverse marine community, this area features a range of fish species, marine invertebrates such as Mollusca, and even prairies of flowering plants like Cymodocea nodosa [26,27,28,29]. In this work, we focused on Miracle and Arrabassada Beaches, located in front of the city of Tarragona, adjacent to the port (Figure 1).

Sediment samples were collected on these two beaches through 2021 and 2022. Each beach underwent two sampling events, with Miracle Beach sampled in June and October 2021 and Arrabassada Beach in February and June 2022. Four collection sites were determined based on the sediment grain size and water column depth, with the first point at 6 m of depth (sand), the second at 13 m (sand), the third at 20 m (transition between sand and silt), and the fourth point at 27 m of depth (silt). Four sub-samples were collected at each point, 15 cm below the seabed surface, using 50 mL plastic tubes. Throughout the collection process, tubes were stored in a refrigerated container and subsequently transported to the laboratory for immediate processing.

### 2.2. Direct Plating, Flocculation Pre-Treatment, Culturing, and Isolation of Fungi

The sediments recovered from each depth (collection sites) were treated separately. However, the sub-samples from each collection site were analyzed together. For this, each set of sub-samples was mixed in a container, vigorously shaken, and then the sediment was poured onto plastic trays with several layers of sterile filter paper to remove excess water. Subsequently, each mixed sediment sample underwent two distinct approaches for fungal isolation: direct plating into sterile Petri dishes (direct plating) and plating after a skimmed milk flocculation (SMF) pre-treatment.

The SMF pre-treatment aimed to improve the probability of detecting microbial content within the collected sediments, following the methodology described by Calgua et al. [18,19,20] and Rusiñol et al. [21]. In summary, 10 g of sediment from each sample was added to 1 L of sterile distilled water, treating each sample in separate containers. A skimmed milk solution was prepared by dissolving 1 g of skimmed milk powder (Difco-Becton, Dickinson and Company, Le Pont de Claix, France) in 100 mL of artificial seawater (33.33 g of sea salts), adjusting the pH to 3.5. Then, 10 mL of this solution was added to each sample to obtain a final concentration of 0.01% of skimmed milk. The samples were stirred for 16 h at room temperature to allow the flocs to settle by gravity. After removing the supernatants, the sediment was collected, transferred to 500 mL centrifuge containers, and centrifuged at 8000× *g* for 30 min at 4 °C. Pellets were suspended in 5 mL of 0.2 M phosphate buffer, pH 7.5 (1:2, *v*/*v* of 0.2 M Na_2_HPO_4_ and 0.2 M NaH_2_PO_4_) and stored at −20 °C.

Moreover, we used three culture media to recover the broadest culturable ascomycete diversity within marine sediments: dichloran rose bengal chloramphenicol agar (DRBC; 5 g peptone, 10 g glucose, 1 g KH_2_PO_4_, 0.5 g MgSO_4_, 25 mg rose bengal, 200 mg chloramphenicol, 2 mg dichloran, 15 g agar, 1 L distilled water), 3% malt extract agar supplemented with sea water (SWMEA3%; 30 g malt extract, 5 g mycological peptone, 15 g agar, 1 L seawater) as a favorable medium for the isolation of marine fungi [31], and potato dextrose agar (PDA; Condalab, Madrid, Spain) supplemented with 2 g/L of cycloheximide (PDA+C) to detect strains resistant to this protein synthesis inhibitor, a frequent trait among various groups of *Ascomycota* [32]. To inhibit bacterial growth, 5 mL of chloramphenicol (15 g/L ethanol) was added to both SWMEA3% and PDA+C culture media.

For direct plating, the procedure involved culturing sediments on SWMEA3% and PDA+C as follows: 1 g of sediment from each sampling point was distributed across two Petri dishes per medium and mixed with the melted medium at 45 °C. In the case of DRBC, only 0.5 g of sediment from each sampling point was distributed across two Petri dishes to address fast-growing fungi. A similar methodology was applied to the sediment pre-treated with SMF, with the exception that 1 mL of floccule from each sampling point was distributed across two Petri dishes in the case of both SWMEA3% and PDA+C, and 0.5 mL of floccule was mixed with DRBC. This step was conducted in duplicate for both approaches. Subsequently, a set of the primary plates from the different culture media was incubated at 22–24 °C, while the other set was incubated at 15 °C to detect fungi capable of growing at lower temperatures. The plates were incubated in darkness and examined under the stereomicroscope for 5–8 weeks.

Pure cultures were obtained on PDA from fragments of the colonies or conidia in the primary plates using a sterile dissection needle. These cultures were used to provide a preliminary morphological identification. Strains that did exhibit sporulation were further subcultured to potato carrot agar (PCA; 20 g potato, 20 g carrot, 15 g agar, 1 L distilled water) or oatmeal agar (OA; 30 g oatmeal, 15 g agar, 1 L distilled water) to promote sporulation.

### 2.3. Morphological Analyses

All strains were carefully examined and compared morphologically with each other for presumed generic identification or species complexes when possible. Due to the large number of strains recovered from the sediments, we only selected a representative subset of strains for molecular identification. In addition, among the strains morphologically identical or similar, a maximum of three strains per collection point were selected for sequencing based on the conserved morphological traits frequently displayed between phylogenetically close ascomycete species [33]. Strains that remained sterile in vitro were also included in the molecular analyses, at the very least, to elucidate their potential affiliation with a fungal group.

For the putative novel species, macroscopic characterization of the colonies was conducted using different culture media, i.e., PDA, MEA (20 g malt extract, 15 g agar, 1 L distilled water), and OA, unless otherwise specified in the species description. Color notations in descriptions are followed by Kornerup and Wanscher [34]. Microscopic characterization was made on OA after 14 days at 25 °C in darkness, unless otherwise specified in the description. Reproductive structures were mounted with lactic acid and observed under an Olympus BH-2 bright-field microscope (Olympus Corporation, Tokyo, Japan). The descriptions were based on a minimum of 30 measurements of the relevant structures to provide size ranges. A Zeiss Axio-Imager M1 light microscope (Zeiss, Oberkochen, Germany) with a DeltaPix Infinity digital camera was used to develop photomicrographs. Photoplates were assembled using Photoshop CS6 v13.0.

The ability of each novel species to grow at different temperatures was also assessed on PDA from 5 to 40 °C at intervals of 5 °C, including measurements at 37 °C.

The strains recovered throughout the study were preserved in the culture collection of the Faculty of Medicine in Reus (FMR, Reus, Catalonia, Spain). In addition, the strains of novel or rare fungi were deposited at the Westerdijk Fungal Biodiversity Institute in Utrecht (CBS, Utrecht, The Netherlands), together with holotypes (i.e., dry colonies on the most appropriate media for their sporulation) and cultures of the selected type strains. The nomenclatural novelties were registered in MycoBank (https://www.mycobank.org/; accessed on 6 February 2024).

### 2.4. DNA Extraction, PCR Amplification, Sequencing, and Strain Identification

The strains selected based on morphology were further delineated through DNA barcodes. Genomic DNA was extracted using the modified protocol of Müller et al. [35] and quantified with a Nanodrop 2000 (Thermo Scientific, Madrid, Spain). Barcoding primers were selected based on the preliminary morphological identification and the most recent phylogenetic studies of the respective groups. The primer pairs and PCR conditions used for the different fungal groups recovered in this study are specified in Table 1. PCR products were purified and sequenced at Macrogen Corp. Europe (Madrid, Spain) using the same primers employed for amplification. Consensus sequences were assembled using SeqMan v. 7.0.0 (DNAStar Lasergene, Madison, WI, USA).

The resulting sequences were compared with those available at the National Center for Biotechnology Information (NCBI) using the Basic Local Alignment Search Tool (BLAST; https://blast.ncbi.nlm.nih.gov/Blast.cgi, accessed on 6 October 2023). A maximum similarity level of >98% with ≥90% sequence coverage was used to determine related GenBank sequences. The accessions of these sequences were carefully sorted to determine those associated with type strains of accepted species. Additional barcodes were sequenced to resolve phylogenetically close taxa. Strains were assigned a species name based on morphology and the percentage of similarity to type strains. Similarity values <98% were considered indicative of potential unknown fungi, and these strains were subjected to further analyses. Each strain was delineated using the barcodes listed in Appendix A).

### 2.5. Phylogenetic Analyses

The strains that could represent potential unknown fungi were further analyzed under a consolidated species concept, combining morphological and phylogenetic data [44]. Alignments were constructed, including the sequences with the closest percentage of identity from the BLAST search and sequences from the most recent phylogenetic studies of the respective groups, namely *Exophiala* [45], *Nigrocephalum* [46], *Queenslandipenidiella* [44], and the family *Schizotheciaceae* [47,48]. Sequences were aligned using the ClustalW algorithm [49] in MEGA (Molecular Evolutionary Genetics Analysis) software v. 6.0 [50] and refined with MUSCLE [51] or adjusted manually as needed. The analyses included more than one genetic region based on the previous phylogenies of each group [44,45,46,47,48]. Each region was aligned individually before being combined into a single dataset. The phylogenetic concordance among phylogenies was visually assessed to identify incongruent results among clades with high statistical support. Following the confirmation of concordance, the separate alignments were concatenated into a single data matrix in MEGA [50].

Maximum likelihood (ML) analyses were conducted using the CIPRES Science Gateway portal v. 3.3 (https://www.phylo.org/, accessed on 5 December 2023; Ref. [52]) and RAxML-HPC2 on ACCESS v. 8.2.12 [53] with the default GTR substitution matrix and 1000 rapid bootstrap replications. Bootstrap support (bs) ≥70 was considered significant [54]. Bayesian analyses were performed using MrBayes v. 3.2.6 [55]. The best substitution model for each locus was estimated using jModelTest v. 2.1.3 following the Akaike criterion [56,57]. Markov chain Monte Carlo sampling (MCMC) was performed for 10 million generations using four simultaneous chains (one cold chain and three heated chains) starting from a random tree topology. Trees were sampled every 1000th generation or until the run was stopped automatically when the average standard deviation of split frequencies fell below 0.01. The first 25% of the trees were discarded as the burn-in phase of each analysis, and the remaining trees were used to calculate posterior probabilities (pp). A pp value of ≥0.95 was considered significant [58]. The resulting trees were plotted using FigTree v. 1.3.1 (http://tree.bio.ed.ac.uk/software/figtree/, accessed on 5 December 2023). The DNA sequences generated in this study were deposited in GenBank (Appendix A), and the alignments of the novel species were submitted to Zenodo (https://doi.org/10.5281/zenodo.10658628, accessed on 20 May 2023). Information about representative strains and outgroups used in the phylogenetic analyses is provided in Appendix A).

### 2.6. Representation of Diversity Data

The relative abundance of the detected diversity was visualized at the family and genus levels to analyze trends in diversity across beaches and collection sites. Strains representing known genera but not assigned to any existing family, as well as unidentified strains due to lack of sporulation or inability to extract DNA for molecular identification, were labeled as *incertae sedis*. We used node networks to represent diversity at the species level using the software Cytoscape v3.10.2 [59]. This way, we could represent the diversity detected at each beach, assess the performance of direct plating and the SMF pre-treatment to detect ascomycetes, and we could also represent the connection between species, culture media, and temperature.

## 3. Results

Among the marine sediments processed through direct plating and the SMF pre-treatment, we recovered 362 strains of filamentous ascomycetes, with 190 strains from Miracle Beach and 172 strains from Arrabassada Beach. Following morphological examination, 188 strains (52%) were selected for sequencing (Appendix A), encompassing all the morphological diversity detected, with the majority identified at the genus level.

### 3.1. Diversity at the Family and Genus Level

The recovered strains represented 29 families of ascomycetes, with *Aspergillaceae* and *Microascaceae* being the most prevalent on both beaches (Figure 2A). The families detected accounted for 54 genera, 42 identified at Arrabassada Beach and 28 at Miracle Beach, with 16 genera detected at both beaches (Figure 2B).

Considering both beaches, the genera most frequently detected were *Aspergillus* (15%), *Penicillium* (14%), and *Scedosporium* (7%). Miracle Beach is characterized by the prevalence of *Aspergillus* (21%) and *Penicillium* (18%), followed by genera like *Talaromyces* (7%), *Queenslandipenidiella* (6%) and *Emericellopsis* (4%), which were detected in lower frequency. In contrast, Arrabassada Beach exhibited more heterogeneity, with *Scedosporium* (12%) being the most frequently detected genus, followed by *Penicillium* (11%), *Aspergillus* (9%), *Talaromyces* (4%), and *Amphichorda* (3%). Regarding the genera exclusively recovered from each of the beaches, *Byssoonygena*, *Cladophialophora*, *Cladosporium*, *Cucurbidothis*, *Exophiala*, *Gymnoascus*, *Parathielavia*, *Pseudogymnoascus*, *Queenslandipenidiella*, *Stolonocarpus*, and *Triadelphia* were detected in Miracle Beach, whereas *Acrophialophora*, *Aphanoascus*, *Arachnomyces*, *Botryotrichum*, *Chaetomium*, *Chantransiopsis*, *Collariella*, *Corollospora*, *Fusarium*, *Gliomastix*, *Gymnoascoideus*, *Lophotrichus*, *Microascus*, *Nigrocephalum*, *Paraphaeosphaeria*, *Parapyrenis*, *Pseudohumicola*, *Roussoella*, *Schizothecium*, *Sporothrix*, *Verruciconidia*, *Waltergamsia*, *Westerdykella*, and *Zopfiella* were exclusively recovered from Arrabassada Beach.

The complete list of detected genera at each depth for both beaches can be accessed in Figure 2C,D. Taxa belonging to *Aspergillus* and *Penicillium* were detected at all collection sites on both beaches. At Miracle Beach (Figure 2C), most of the diversity was observed at depths of 20 and 27 m. Fifteen genera were exclusively recovered from these two collection sites, including *Amphichorda* and *Pseudeurotium*, which were exclusively isolated at 20 m of depth, and *Parasarocladium* and *Pseudogymnoascus* exclusively at 27 m of depth. At Arrabassada Beach (Figure 2D), *Amphichorda* and *Pseudeurotium* were recovered at both 20 and 27 m, while *Parasarocladium* was exclusively isolated again at 27 m of depth. Twelve genera were exclusively recovered from 27 m depth in this beach, including *Schizothecium* and *Westerdykella*, which were absent in Miracle Beach.

It is noteworthy that, on both beaches, the diversity detected increased with the depth of collection, particularly below 13 m. In fact, genera recovered from depths of 20 and 27 m represented approximately 60% of the total diversity.

### 3.2. Diversity at the Species Level

#### 3.2.1. Species Detected in Miracle and Arrabassada Beaches

The strains selected for molecular analyses represented 94 species, 60 from Arrabassada Beach and 54 from Miracle Beach, with 20 species detected at both beaches. The complete list of species can be accessed in Figure 3.

Among these species, 21 could not be assigned to any extant taxa and require further analyses to ensure their correct identification. However, in the present study, we have resolved the taxonomy of four of them as putative new species. The strains FMR 19606 and FMR 19607 were morphologically identified as members of the genus *Exophiala* and likely belonging to the *jeanselmei*-clade due to their subcylindrical, pale olivaceous, annellidic conidiogenous cells.

The strains FMR 19852, FMR 20069, and FMR 20174 were recognized as belonging to *Nigrocephalum* based on their dark-pigmented colonies, subcylindrical and phialidic conidiogenous cells, with conspicuous collarettes, producing conidia in dark-pigmented slimy heads.

The set of nine marine strains (FMR 19473–FMR 19481) was identified as similar to the genus *Queenslandipenidiella* based on their brown and macronematous conidiophores with polyblastic conidiogenous cells that give rise to at least two sets of ramoconidia.

On the contrary, the strain FMR 20114 could not be recognized as any known genus since it only produced intercalary chlamydospore-like cells.

Since a BLAST search using different barcoding regions of these strains could not assign them to any known species of the mentioned genera, we proceeded to their phenotypic and genetic characterization. The phylogenetic delineation and morphological description of these new taxa are provided below, and they are proposed as *Exophiala* (*E*.) *littoralis*, *Nigrocephalum* (*N*.) *paracollariferum*, *Queenslandipenidiella* (*Q*.) *verrucosa*, and *Schizochlamydosporiella* (*S*.) *marina* in the taxonomy section.

#### 3.2.2. Direct Plating and SMF Pre-Treatment: Effect of Culture Media and Temperature

To assess the performance of direct plating vs. SMF pre-treatment on diversity, the species detected at both beaches were analyzed in combination. As a result, 70 species were detected through direct plating and 42 species through SMF, with 17 species being detected through both approaches. The species exclusively detected by each approach are shown in Figure 4. Of note is that the flocculation pre-treatment enabled the isolation of 20 species that were not obtained through direct plating, increasing the number of species detected by 25%.

In addition to direct plating and SMF, we used three different culture media incubated at two temperatures to further increase the chance of detecting filamentous ascomycetes. The complete list of species detected through direct plating and SMF considering the medium of isolation can be accessed in Figure 4A. A total of 61 species were detected with SWMEA3%, 40 with DRBC, and 19 with PDA+C. Species like *Penicillium antarcticum*, *Q. verrucosa,* and *Stachybotrys chlorohalonatus* were isolated in the three media, with up to 19 species detected in more than one medium. Around 90% of the diversity was recovered with SWMEA3% and DRBC.

The medium that enabled the recovery of most of the diversity was SWMEA3%, with 41 species obtained exclusively from it. DRBC enabled the detection of different fungal species that overlapped with SWMEA3% by only 30%. Species from the genus *Talaromyces* were exclusively detected through DRBC. The medium PDA+C provided the least number of species. However, this was the only medium that enabled the recovery of onygenalean fungi like *Aphanoascus crassitunicatus*, *Gymnoascoideus* sp., *Gymnoascus longitrichus*, *Malbranchea zuffiana,* and *Narasimhella poonensis*. Regarding the effect of temperature (Figure 4B), a total of 69 species were detected at 15 °C and 59 species at 25 °C. Of these, 37 were exclusively detected at 15 °C, and 27 were exclusively detected at 25 °C.

### 3.3. Phylogeny

The unidentified strains of *Exophiala* (FMR 19606 and FMR 19607), *Nigrocephalum* (FMR 19852, FMR 20069, and FMR 20174), *Queenslandipenidiella* (FMR 19473–FMR 19481), and the strain FMR 20114 that could not be assigned to any known genera of *Ascomycota* were subjected to phylogenetic analyses with different gene markers depending on the fungal group to which they were related. The number of conserved, variable, and parsimony informative sites for each alignment, together with the models used in each analysis, are shown in Table 2.

A BLAST search using the ITS and LSU regions confirmed the relationship of the *Exophiala* marine strains with the *jeanselmei*-clade, showing a 99% sequence similarity with LSU sequences regarding those of the reference strain of *E. oligosperma* (CBS 725.88). However, for the ITS region, the maximum percentage of similarity was around 90% with the species *E. lamphunensis* (CMU 404) and *E. saxicola* (CMU 415). For species delineation, phylogenetic analyses were conducted using ITS and LSU sequences of the species of the *jeanselmei*-clade. Sequences of each gene marker were aligned individually (Appendix A), and after confirming the absence of incongruences, these sequences were concatenated into a single matrix. The final alignment comprised 27 sequences representing ex-type and reference strains of the species of the *jeanselmei*-clade and the outgroups selected (*E. dehoogii* CBS 149779 and *E. palmae* UPCB 86822). The resulting phylogenetic tree (Figure 5) resolved the unidentified strains of *Exophiala* in an independent and well-supported lineage separated from the other species of the *jeanselmei*-clade, which represents a novel species for the genus *Exophiala*.

The marine strains of *Nigrocephalum* were also analyzed through a BLAST search using the ITS and LSU regions. The identification at the genus level was confirmed through the LSU region with a 99% similarity with the ex-type strain of *N. collariferum* (CBS 124586), the type species of the genus *Nigrocephalum*. The ITS sequences showed a 97% similarity with the same strain. For species delineation, we performed phylogenetic analyses with sequences of the ITS, LSU, *tef*1-α, and *rpb*2 loci. Each locus was aligned individually, and the resulting tree topologies were similar and without incongruences (Appendix A). The alignments were, therefore, concatenated into a single matrix. The final alignment included 20 strains that comprised the marine strains, the type species of *Nigrocephalum*, and ex-type or reference strains of species of other genera of *Plectosphaerellaceae* related to *Nigrocephalum*, as well as the selected outgroups for the analyses (*Phialoparvum bifurcatum* CBS 299.70B and *Plectosphaerella cucumerina* CBS 137.33). In the phylogenetic tree (Figure 6), the marine strains of *Nigrocephalum* formed an independent and fully supported sister clade with that representative of *N. collariferum*, but with enough phylogenetic distance (99% LSU, 97% ITS, 97% for *tef*1-α, and 90% for *rpb*2) to be considered a new species for the genus *Nigrocephalum*.

The BLAST search of the strains morphologically identified as *Queenslandipenidiella* with LSU and ITS sequences confirmed their placement into the family *Teratosphaeriaceae* and showed a similarity of around 96% with *Q. kurandae* (CBS 121715) using the LSU region and around 88% with the same strain using the ITS barcode. While the strains FMR 19473–FMR 19477 and 19481 were genetically identical, the strains FMR 19478 and FMR 19479 showed slight genetic variability. Therefore, only the strains FMR 19477, FMR 19478, and FMR 19479 were included in the phylogenetic analyses with the ITS and LSU regions to prevent branch imbalance in the resulting phylogenetic tree. Since separate alignments of each region and the resulting tree did not show incongruences (Appendix A), they were concatenated into a single matrix. The final alignment comprised 29 strains representing genera in the *Teratosphaeriaceae* and the outgroups *Ramularia* (*R*.) *eucalypti* (CBS 120726) and *R. endophylla* (CBS 113265). The resulting phylogenetic tree resolved the marine strains as an independent lineage, which was considered representative of a new species of the genus *Queenslandipenidiella* (Figure 7).

Finally, a BLAST search with ITS and LSU sequences revealed that the marine strain FMR 20114 was related to members of the family *Schizotheciaceae*. The highest percentage of similarity was obtained with sequences of the ex-type strains *Apiosordaria* (*A*.) *microcarpa* (CBS 692.82: 95% ITS and 98% LSU) and *Apodus deciduus* (CBS 506.70: 90% ITS and 98% LSU). For a more precise identification, in addition to the phylogenetic analyses with ITS and LSU, we also analyzed the sequences of the *rpb*2 gene. Once the lack of incongruences was confirmed (Appendix A), individual alignments were concatenated into a single matrix. The final alignment comprised 29 strains, which, in addition to the unidentified strains, included ex-type and reference strains of the species belonging to the *Schizotheciaceae* family and the selected outgroups (*Amesia atrobrunnea* CBS 379.66 and *Triangularia bambusae* CBS 352.33). The resulting phylogenetic tree resolved the strain FMR 20114 as an independent lineage within the *Schizotheciaceae* (Appendix A). However, due to the great phylogenetic distance between different genera and the lack of ITS or *rpb*2 sequences for numerous species in the family (Appendix A), we performed an additional phylogenetic analysis only with the strains of the species phylogenetically close to FMR 20114 and with the three gene markers. The resulting tree (Figure 8) resolved the strain FMR 20114 in a singleton distant branch, phylogenetically distant from its counterpart *A. microcarpa* (98% LSU, 95% ITS, 81% *tub*2, 77% *rpb*2), suggesting an undescribed genus for the family that is proposed as *Schizochlamydosporiella* in the taxonomy section.

### 3.4. Taxonomy

***Exophiala littoralis*** Guerra-Mateo, Cano & Gené, sp. nov. Figure 9.

MycoBank 852016

*Etymology*. The name refers to the area where the species was collected, the western Mediterranean coast.

*Type*. Spain, Catalonia, Mediterranean coast, Tarragona, Platja del Miracle, 41°6′19″ N, 1°15′37″ E, from sediments at 6 m depth, June 2021, *G. Quiroga-Jofre* and *D. Guerra-Mateo* (holotype CBS H-25351, ex-type FMR 19606, CBS 151312).

*Classification*. *Eurotiomycetes*, *Chaetothyriales*, *Herpotrichiellaceae.*

*Saprobic* on marine sediments at 6 m of depth. *Asexual morph* on OA at 25 °C. *Mycelium* is composed of septate, branched, smooth- and thick-walled, pale olivaceous to brown, 2–3 µm wide hyphae; sparse torulose hyphae are present on PDA. *Conidiophores* micronematous are often reduced to conidiogenous cells growing directly from vegetative hyphae. *Conidiogenous cells* are annellidic, mostly lateral, arising in acute angles, terminal and intercalary also present, cylindrical, tapering terminally in an irregular annellated zone, pale olivaceous to brown, (6–)10–20(–26) × 1.5–3 µm. *Conidia* are one-celled, smooth-walled, pale olivaceous, subcylindrical to clavate, 3–6 × 1.5–2 µm, often with a truncate and prominent basal scar. Budding cells were not observed. *Sexual morph* is unknown.

*Culture characteristics* (after 14 days at 25 °C). Colonies on the PDA attaining 27–28 mm diam., umbonate, felty, dark green (30F6) at center to grayish green (30E5) towards the periphery, margin entire; reverse dark green (30F4–30F6). On MEA, reaching 26–27 mm diam., umbonate, floccose, pale gray (1B1) at the center to brownish orange (5C5) towards the periphery, margin entire; reverse grayish brown (5E3) to brownish orange (5C5) at the periphery. On OA, attaining 22 mm diam., slightly umbonate, floccose turning velvety towards the periphery, dark gray (1E1), margin entire; reverse dark gray (1E1). The diffusible pigment was not observed on any medium.

*Cardinal temperatures for growth*. Minimum 5 °C (2 mm), optimum 30 °C (30 mm), maximum 37 °C (2 mm).

*Additional specimens were examined*. Spain, Catalonia, Mediterranean coast, Tarragona, Platja del Miracle, 41°6′19″ N, 1°15′37″ E, from sediments at 6 m depth, June 2021, *G. Quiroga-Jofre* and *D. Guerra-Mateo* (FMR 19607).

*Notes*. The genus *Exophiala* represents a polyphyletic group in the family *Herpotrichiellaceae*, where *E. littoralis* is phylogenetically placed in the so-called *jeanselmei*-clade [45,60]. This species is morphologically similar to other species of this clade, such as *E. hongkongensis*, *E. jeanselmei*, *E. nishimurae*, *E. oligosperma,* and *E. pseudooligosperma*, by the production of cylindrical conidiogenous cells (Figure 5; Refs. [61,62,63,64,65,66]). However, the production of claviform conidia with thick and truncate bases, the absence of budding cells, and the phylogenetic distance with the ITS/LSU barcodes were clear features to distinguish *E. littoralis* from the rest of the members of the *jeanselmei*-clade.

***Nigrocephalum paracollariferum*** Guerra-Mateo, Cano & Gené, sp. nov. Figure 10.

MycoBank 852017

*Etymology*. The name refers to the morphological similarity and phylogenetic relationship with the type species of the genus *N. collariferum.*

*Type*. Spain, Catalonia, Mediterranean coast, Tarragona, Platja de la Arrabassada, 41°6′53″ N, 1°16′48″ E, from sediments at 20 m depth, March 2022, *G. Quiroga-Jofre* and *D. Guerra-Mateo* (holotype CBS H-25352, ex-type FMR 20069, CBS 151313).

*Classification*. *Sordariomycetes*, *Glomerellales*, *Plectosphaerellaceae*.

*Saprobic* on marine sediments at 6, 13, and 20 m depth. *Asexual morph* on OA at 25 °C. *Mycelium* is composed of septate, branched, finely to roughly warted, thick-walled, pale olive to brown, 2–3 μm wide hyphae. *Conidiophores* micronematous, usually consisting of single phialides on slightly darker subtending cells, occasionally basitonously branched, arising in acute angles from vegetative hyphae, smooth-walled, pale olive-brown to dark brown. *Phialides* are mostly lateral, subcylindrical to subulate, wavy at the apex, with funnel-shaped collarettes and conspicuous periclinal thickening, pale olive-brown, 30–46 × 1.5–2.5 µm. *Conidia* are one-celled, thick- and smooth-walled, pale brown, ellipsoidal, 3.8–4.5 × 2.3–3 µm, arranged in slimy dark heads. *Chlamydospores* were not observed. *Sexual morph* is unknown.

*Culture characteristics* (after 14 days at 25 °C). Colonies on PDA, reaching 10–12 mm diam., crateriform, radially sulcate, velvety, brownish gray (5E2), margin irregularly lobated; reverse light gray (5D1). On MEA, reaching 4–5 mm diam., flat, gray (5F1) at the center to brownish gray (5E2) and yellowish white (4A2) at the periphery, margin entire, slightly lobated; reverse concolorous. On OA, attaining 22–24 mm diam., flat, brownish gray (5E2) to white (1A1) at the periphery, margin diffusing in the medium. The diffusible pigment was not observed on any medium.

*Cardinal temperatures for growth*. Minimum 10 °C (4 mm), optimum 25 °C (10–12 mm), maximum 35 °C (2 mm).

*Additional specimens were examined*. Spain, Catalonia, Mediterranean coast, Tarragona, Platja de la Arrabassada, 41°7′3″ N, 1°16′42″ E, from sediments at 6 m depth, March 2022, *G. Quiroga-Jofre* and *D. Guerra-Mateo* (FMR 19852); ibid., N 41°6′57″ N, E 1°16′45″ E, from sediments at 13 m depth, June 2022, *G. Quiroga-Jofre* and *D. Guerra-Mateo* (FMR 20174).

*Notes*. The genus *Nigrocephalum* was erected to accommodate a dark-pigmented *Acremonium* species, *A. collariferum*, within the family *Plectosphaerellaceae* [46]. The genus is monotypic, with *N. collariferum* isolated from human toenails. *Nigrocephalum paracollariferum* is morphologically similar to the type species; it differs only in the completely ellipsoidal conidia and the absence of chlamydospores in any of the culture media tested, instead of the reniform in lateral view conidia and intercalary chlamydospores of the type species [67].

***Queenslandipenidiella verrucosa*** Guerra-Mateo, Cano & Gené, sp. nov. Figure 11.

Mycobank MB 848254

*Etymology*. The name refers to the verrucose ornamentation of the conidiophore.

*Type*. Spain, Catalonia, Mediterranean coast, Tarragona, Platja del Miracle, 41°6′19″ N, 1°15′37″ E, from sediments at 13 m depth, June 2021, *G. Quiroga-Jofre* and *D. Guerra-Mateo* (holotype CBS H-25258, ex-type FMR 19477, CBS 149939).

*Classification*. *Dothideomycetes*, *Mycosphaerellales*, *Teratosphaeriaceae*.

*Saprobic* on marine sediments at 6, 13, and 20 m depth. *Asexual morph* on OA at 25 °C. *Mycelium* is composed of septate, branched, smooth to verrucose, thick-walled, brown, 2–3 µm wide hyphae. *Conidiophores* semi-macronematous to macronematous, arising directly from superficial mycelium, erect, unbranched, occasionally branched, brown, verrucose, thick-walled, up to 110 µm long; stipe up to 5-septate, cells 10–25 × 2–3 µm. *Conidiogenous cells* are integrated, terminal or intercalary, verrucose, thick-walled, brown, cylindrical to subcylindrical, 10–25 × 2–4.5 µm, often slightly inflated terminally, polyblastic, proliferating sympodially at the apex, showing 2–4 conidiogenous loci, slightly denticulate and pigmented, giving rise to at least two sets of ramoconidia. *Ramoconidia* with up to 6 distal conidiogenous loci, finely roughened, thick-walled, brown, subcylindrical to cylindrical-oblong, 3.5–7(–8) × 3–4 µm, become sinuous and tortuous with age. *Conidia* solitary or arranged in short chains, smooth- to finely rough-walled, brown, subglobose to obpyriform, 3–4.5 × 2–4 µm. *Sexual morph* is not observed.

*Culture characteristics* (after 14 days at 25 °C). Colonies on MEA 2%, attaining 8–10 mm diam. (16–19 mm after 28 days), circular, felty, slightly raised, greenish gray (30D2) to grayish green (30D6) and dark green (30F8) at the periphery, margin regular and feathery; reverse gray (30F1) to dark green (30F8) at the periphery. On OA, attaining 2 mm diam. (3–4 mm after 28 days), raised, irregular, with abundant aerial mycelium, margin feathery, pastel gray (30D1); reverse gray (30F1). Sporulation is abundant in all culture media. The diffusible pigment was not observed on any medium.

*Cardinal temperatures for growth*. Minimum 15 °C (3 mm), optimum 25 °C (10 mm), maximum 35 °C (2 mm).

*Additional specimens examined*: Spain, Catalonia, Mediterranean coast, Tarragona, Platja del Miracle, 41°6′19″ N, 1°15′37″ E, from sediments at 6 m depth, June 2021, *G. Quiroga-Jofre* and *D. Guerra-Mateo* (FMR 19481); ibid., from sediments at 13 m depth (FMR 19478, FMR 19479, 19480); ibid., from sediments at 20 m depth (FMR 19473, FMR 19474, FMR 19475, FMR 19476).

*Notes.* The genus *Queenslandipenidiella* was erected to accommodate *Penidiella kurandae* based on the phylogenetic distance and distinct morphological features regarding *P. columbiana*, the type species of the genus *Penidiella* [44,68]. *Queenslandipenidiella* verrucosa can be primarily distinguished from *Q. kurandae* based on conidiophore morphological traits, which are verrucose and commonly unbranched in the former species and smooth and penicillate-branched in *Q. kurandae* [44,68]. It is worth mentioning that *Q. verrucosa* shows a considerable phylogenetic distance with respect to *Q. kurandae*. However, the phylogenetic distance between these two species for each gene marker is within the range of interspecific variability accepted in *Teratosphaeriaceae* (ITS 90%, LSU 97%, *tef*1-α 89%, *tub*2 88%; Ref. [44]). This classification, however, may be subject to taxonomical changes with further studies on the biodiversity and phylogeny of the family.

***Schizochlamydosporiella*** Guerra-Mateo, Gené & Cano, gen. nov.

MycoBank 852018

*Etymology*. The name refers to the production of chlamydospores released by schizolythic secession over time.

*Classification*. *Sordariomycetes*, *Sordariales*, *Schizotheciaceae*.

*Mycelium* is superficial and immersed, septate, torulose, smooth- and thin-walled, hyaline to pale brown, with hyphae with intercalary chlamydospore-like cells. *Chlamydospore-like cells* are single or in chains, sometimes with several branching loci, one- or two-celled, smooth- and thick-walled, brown, subglobose to subcylindrical, ellipsoidal, barrel-shaped, or irregularly shaped, with terminal, dark pigmented, thick-walled scars, released from the chain by schizolythic secession. *Sexual morph* is unknown.

*Type species*. *Schizochlamydosporiella marina* Guerra-Mateo, Gené & Cano.

***Schizochlamydosporiella marina*** Guerra-Mateo, Gené & Cano, sp. nov. Figure 12.

MycoBank 852019

*Etymology.* The name refers to the habitat where the type strain was isolated, marine sediments.

*Type*. Spain, Catalonia, Mediterranean coast, Tarragona, Platja de la Arrabassada, 41°6′45″ N, 1°16′51″ E, from sediments at 27 m depth, March 2022, *G. Quiroga-Jofre* and *D. Guerra-Mateo* (holotype CBS H-25353, ex-type FMR 20114, CBS 151315).

*Saprobic* on marine sediments at 27 m of depth. *Asexual morph* on OA at 25 °C. *Mycelium is* composed of superficial and immersed, septate, torulose, smooth- and thin-walled, hyaline to pale brown, 1–2 µm wide hyphae with intercalary chlamydospore-like cells. *Chlamydospore-like cells* are single or in chains; single chlamydospore cells, one-celled, rarely two-celled and constricted at septum, smooth- and thick-walled, brown, subglobose to subcylindrical, ellipsoidal, barrel-shaped or irregularly shaped, (8–)10–15(–17) × 3–6 µm, with terminal, dark pigmented and thick-walled scars; chains reaching up to 190 µm in length, occasionally branched, with up to three branching loci, cells released from the chain by schizolythic secession in old cultures. *Sexual morph* is unknown.

*Culture characteristics* (after 28 days at 25 °C). Colonies on MEA reaching 32–36 mm diam., slightly raised, radially sulcate underneath the aerial mycelium, cottony, golden gray (4C2) to dark gray (1F1) and yellowish gray (4B2) at periphery, margin entire and lobated; reverse dark gray (1F1) to golden gray (4C2). On OA, reaching 85 mm diam., velvety with short aerial mycelium that becomes prominent towards the margin, brownish gray (4D2) at center with scattered patches of dark gray (1F1), margin entire; reverse dark gray (1F1). On CMA, reaching 56–60 mm diam., slightly raised, radially sulcate, velvety, golden gray (4C2) at the center to dark gray (1F1) and olive brown (4D3) at the periphery, margin entire and lobated; reverse dark gray (1F1). On PCA, reaching 65–70 mm diam., predominantly submerged with short and sparse aerial mycelium, dark gray (1D1) to brownish gray towards the margin (4D2), margin feathery; reverse dark gray (1F1).

*Cardinal temperatures for growth*. Minimum 5 °C (2 mm), optimum 30 °C (56 mm), maximum 40 °C (2 mm).

*Notes*. *Schizochlamydosporiella marina* represents an independent lineage within the *Schizotheciaceae* (Figure 8 and Appendix A). This family was erected to resolve an independent lineage in *Lasiosphaeriaceae s.l.* that comprises the genus *Schizothecium*, together with many species whose identification needs a taxonomical revaluation [47,48]. The *Schizotheciaceae* were circumscribed based on the morphological characters of the sexual morph, which was not produced by our species despite testing different culture media, even mixed with plant material (leaves of *Chamaerops humilis* and *Cymodocea nodosa*), and up to two months of incubation. *Schizochlamydosporiella marina* only produces brown chlamydospore-like cells that are released by schizolythic secession. By definition, chlamydospores represent resting spores that are not dispersed [69]. Thus, the cells produced by *S. marina* suggest a kind of dispersal asexual structure like conidia, but we could not observe in vitro germination of the cells. Of note is that this particular type of asexual morph has not been reported before in the family. The asexual morphs in *Schizotheciaceae* are commonly displayed as either phialidic conidiogenous cells that produce ovoid to clavate, hyaline to pale brown conidia, or holoblastic conidia in a chrysosporium-like fashion, like in *A. microcarpa* [47,70]. We propose *S. marina* as a type species of the novel genus *Schizochlamydosporiella* based on the great phylogenetic distance with its closest *species, A. microcarpa* (CBS 692.82) and unique morphology in comparison to other members of the family. However, the morphological description of our fungus is clearly susceptible to emendation when new isolates of the species become available.

## 4. Discussion

This work aimed to detect the diversity of ascomycetous taxa from marine sediments in the Miracle and Arrabassada Beaches, western Mediterranean Sea. For this, we combined the use of direct plating and an SMF pre-treatment with three culture media and two incubation temperatures. This methodology enabled the detection of 94 species, which represent a diverse fungal community that exceeds the number of species typically detected in culture-dependent studies in the marine environment [71,72,73,74]. The combination of both approaches improved our ability to detect fungi in marine sediments, including rare and novel species. In fact, although most of the species were detected through direct plating, up to 20 species were exclusively recovered with SMF (Figure 4).

### 4.1. Effect of In Vitro Conditions for Isolation of Marine Fungi

Conventional methods for studying marine fungi include dilution plating, membrane filtering, baiting, single spore isolation, and the deployment of wood or man-made panels in the sea [14,17]. This study represents the first time that SMF has been employed for the detection of marine fungi.

Our results show that, independently of the media and temperatures used for isolation, we detected different communities with direct plating and SMF. It is noteworthy that the SMF-pre-treated sediment was stored at −20 °C, and despite this, fungal propagules were still viable and recovered in synthetic media. This suggests that SMF could be used as a method to store fungi over time. Although the extent of fungal viability under this treatment should be tested experimentally in future research, this opens the door for a more efficient study of the hard-to-reach marine samples. For these reasons, we recommend the use of SMF in surveys of marine fungi. Particularly in combination with direct plating, enabling the in-depth study of the sample.

Culture media restricts the diversity of detectable fungi. In this regard, we used three isolation media, but SWMEA3% proved to be the most successful one (Figure 4), consistent with the results obtained by other authors [31]. However, to the extent of our knowledge, this medium has never been used in diversity studies. In contrast, DRBC represents a typical medium for the isolation of marine fungi [11,13,73,74]. This medium enabled the isolation of another great portion of species and, most notably, the detection of six *Talaromyces* species that were not obtained with any other medium (Figure 4). In addition, we used a medium supplemented with cycloheximide (PDA+C). Although its use in diversity studies is rare due to the toxicity of this antimicrobial drug to eukaryotes, many marine fungi are resistant to this protein synthesis inhibitor [32]. Therefore, this medium provided the smallest pool of diversity. However, it enabled the isolation of some onygenalean fungi that are likely undescribed ascomycetes (e.g., *Gymnoascoideus* sp., *Malbranchea* sp. 1–3, among others).

In addition, the combination of different temperatures to improve the detection of culturable fungi has been successful in marine habitats [71]. In this case, we combined two incubation temperatures, 25 and 15 °C. We recovered more species from 15 °C; however, this result may be biased by our ability to detect fungal colonies in primary plates. In general, fungi grow more slowly on plates incubated at 15 °C than at 25 °C, thus enabling better detection of single colonies before fast-growing fungi colonize the surface of the plate.

### 4.2. Diversity and Insights of the Recovered Species

In both the Miracle and Arrabassada Beaches, the diversity and the number of ascomycetes that we recovered increased proportionately with the depth of the collection. This trend was particularly evident below 13 m, reaching the maximum diversity at 27–30 m of depth (Figure 2). However, this observation is biased by the small scale of our work in terms of depth. Depth seems to be one of the main factors determining the geographical structure of marine fungi, resulting in an inverse relationship with fungal diversity [75,76,77]. This raises the question: what explains an increase in fungal diversity around 30 m of depth? The answer may be a combination of different environmental factors like temperature, hydrostatic pressure, light availability and its effect on primary production, and the flux of organic matter from the euphotic zone to the deep sediment [71,75,76,77].

The fungal community recovered in this study was dominated by members of the genera *Aspergillus* and *Penicillium*. This trend seems to be constant in surveys of marine fungal diversity worldwide [78,79] and in the Mediterranean Sea [12]. Arrabassada Beach was characterized by the prevalence of the genus *Scedosporium*, with the species *S. apiospermum* and *S. boydii*. Although little is known about this genus in the marine environment, novel species are being described from marine sediments [80]. Other ascomycetes that are typically recovered in sediments of the Mediterranean Sea comprise the genera *Cladosporium* and *Emericellopsis* [11,12,13]. Although in our case only *Cladosporium* (*C*.) *cladosporioides* was isolated, four species of *Emericellopsis* (*Em.*) were recovered (i.e., *Em*. *maritima*, *Em. microspora*, *Em. minima*, and *Em. salmosynnemata*), with *Em. Maritima* and *Em. Salmosynnemata* detected at 6 and 20 m depth in both sampled areas. In addition, we recovered some rare species that represent the first reports for the marine environment, such as *Byssoonygena ceratinophyla*, for which only the ex-type strain is known [81]. We identified the strain FMR 19986 as *Chantransiopsis* (*Ch.*) *c.f. decumbens* based on its morphological resemblance to the protologue of *Ch. decumbens* [82], but with slightly smaller conidia. This species belongs to the *Laboulbeniales* and is supposed to represent an obligate pathogen of arthropods [83], but the marine strain grows perfectly in vitro. Of note is that only SSU sequences for this fungus were available for comparison, and although our strain showed a similarity of 99% with *Ch. decumbens* (LG589), its molecular identification could not be confirmed. Further phylogenetic studies are needed to resolve the taxonomy of the genus *Chantransiopsis* and allied genera.

On the other hand, it is noteworthy the high prevalence of the novel species *Q. verrucosa* in marine sediments. In the sampled areas described in this study, we have isolated multiple strains in sediments ranging in depth from 6 to 20 m. However, we have also detected this species in other areas along Tarragona’s golden coast and the Ebro River Delta (unpublished data), indicating that it is a well-adapted species to the marine habitat. The genus *Queenslandipenidiella* is classified in *Teratosphaeriaceae*, a family of melanized fungi that predominantly inhabit plant material, while others are saprobes that can colonize extreme environments like rock surfaces and even marine habitats [44,84]. The rest of the novel species reported in this study were recovered in less abundance. The ecological knowledge of the genus *Nigrocephalum* is particularly limited. Regarding the new genus *Schizochlamydosporiella*, it is surprising the broad temperature range for growth (5–40 °C) of the new species *S. marina* found at 27 m of depth, although this property is coherent with certain fungi in *Sordariales* [85].

There is evidence of fungal activity throughout the marine environment [86]. In marine food webs, fungi can appear as saprobes, mutualists, endosymbionts, and parasites [87]. Ascomycetes represent active contributors in carbon cycling, degrading pelagic carbohydrates, and reallocating carbon to sediments [15,78,86]. Among the ascomycetes that we have detected in this area, the genera *Aspergillus* and *Penicillium*, together with species like *C. cladosporioides*, *Em. minima*, and *Fusarium solani*, have been reported to degrade different types of plastic polymers [88]. Another species, *Amphichorda* (*A*.) *littoralis*, has been recovered from both sediments and floating rubber, suggesting a role in plastic degradation [89]. Some marine fungi are known to degrade environmental pollutants, like crude oil [90,91,92,93,94]. We have detected the species *E. xenobiotica* and the novel species *E. littoralis* exclusively in sediments from Miracle Beach, adjacent to the port of Tarragona (Figure 1). Some species in *Exophiala* represent extremotolerant colonizers of polluted habitats and nutrient-poor substrates like hydrocarbons [95,96,97]. Moreover, these two species are members of the *Exophiala jeanselmei*-clade, which includes agents of chromoblastomycosis and mycetoma in humans, and like *E. littoralis*, its members can grow at 37 °C [60,98,99,100]. These two *Exophiala* species, as well as *A. littoralis,* represent interesting candidates for future research on their metabolic activities.

Overall, the fungal community detected in this work seems to represent indigenous taxa of the marine environment. This idea is supported by the proximity to a commercial port and the high prevalence of microplastics and other pollutants in submerged sediments from the coast of Tarragona city (Miracle and Arrabassada Beaches; Ref. [25]).

## 5. Conclusions

SMF was successfully employed for the first time to detect fungi in marine sediments and to store them over time. The combination of direct plating and SMF pre-treatment, along with three media and two isolation temperatures, resulted in the detection of 94 species of filamentous ascomycetes. These species represent indigenous taxa of the marine environment, including novel species such as *E. littoralis*, *N. paracollariferum*, *Q. verrucosa,* and *S. marina*.

## Figures and Tables

**Figure 1 jof-10-00281-f001:**
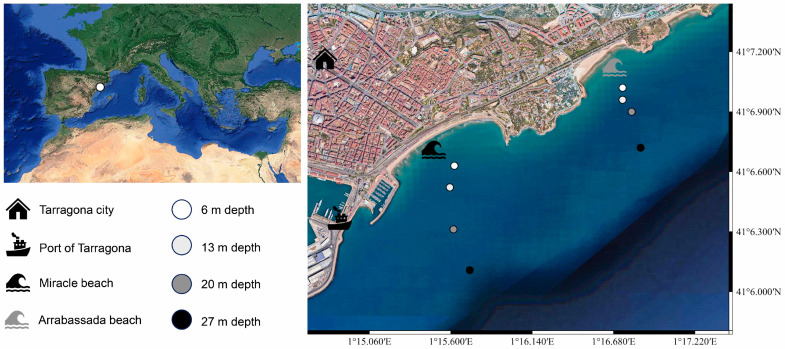
Collection sites at the golden coast of Tarragona, western Mediterranean Sea (Miracle and Arrabassada Beaches, Catalonia, Spain). Map was created using QGIS (v3.34.5; Ref. [30]).

**Figure 2 jof-10-00281-f002:**
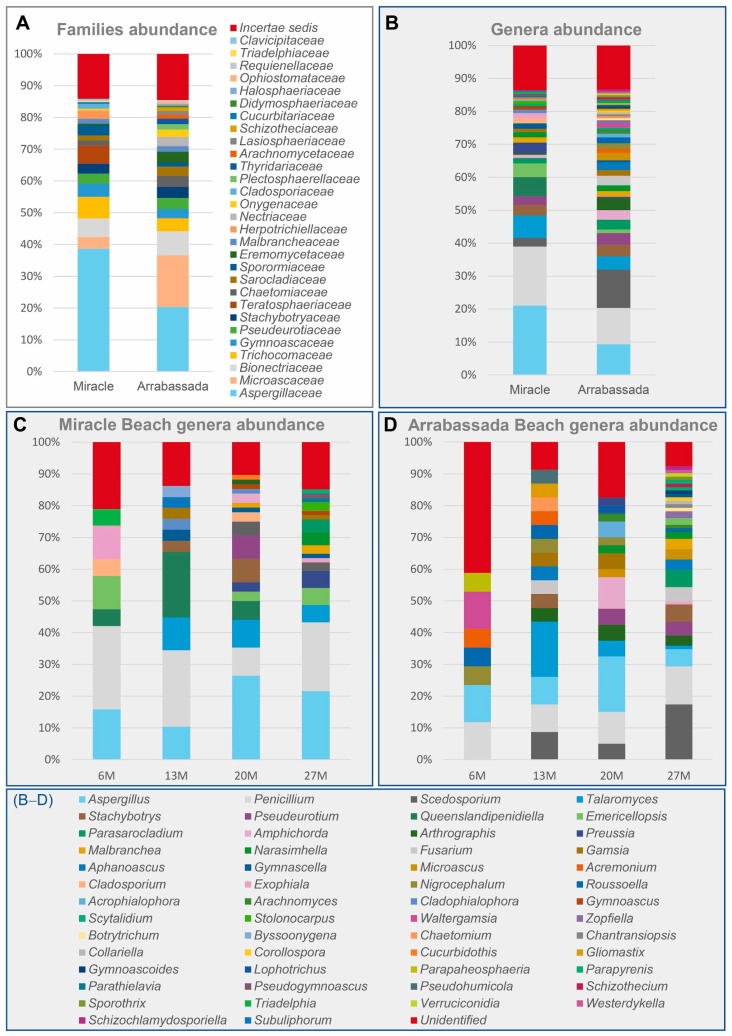
Relative abundance of filamentous ascomycetous taxa was detected in marine sediments from Miracle and Arrabassada Beaches. (**A**) Diversity is detected at each beach at the family level. (**B**) Diversity is detected at each beach at the genus level. (**C**) Genera detected at Miracle Beach, sorted by the depth of the collection point. (**D**) Genera detected at Arrabassada Beach, sorted by the depth of the collection point.

**Figure 3 jof-10-00281-f003:**
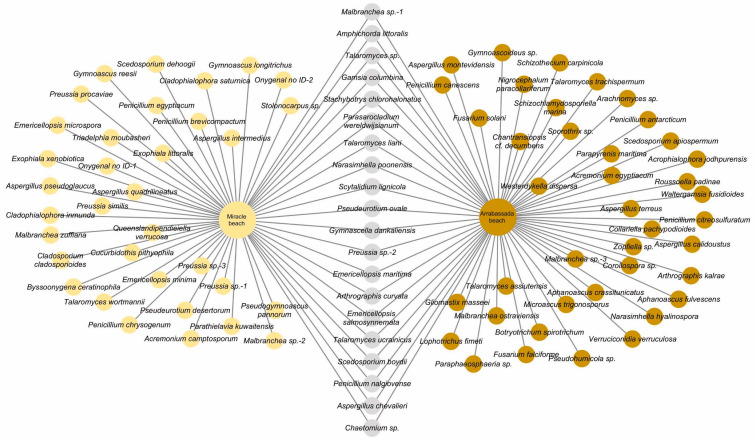
Network representing the species of filamentous ascomycetes detected in marine sediments from Miracle (cream circles) and Arrabassada beaches (brown circles). Grey circles represent species recovered from both areas.

**Figure 4 jof-10-00281-f004:**
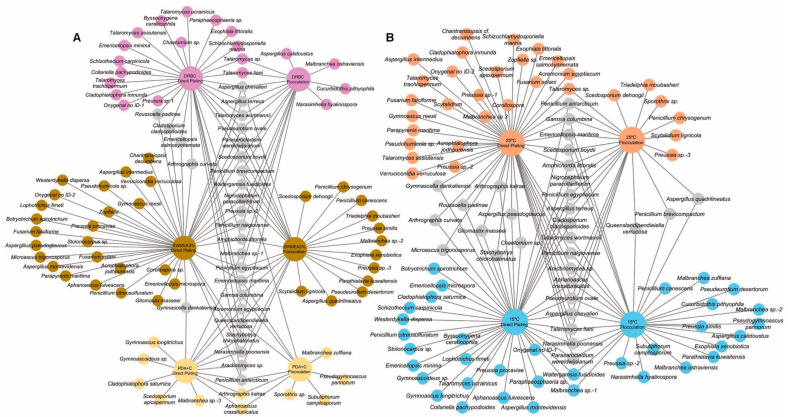
Network representing the species of filamentous ascomycetes detected in marine sediments from Miracle and Arrabassada Beaches through direct plating and flocculation pre-treatment sorted by (**A**) culture media and (**B**) temperature. Grey circles represent species recovered from more than one medium or at both temperatures. The species exclusively recovered from either DRBC, SWMEA3%, PDA+A, 25 °C or 15 °C are highlighted in pink, brown, cream, orange and blue circles, respectively.

**Figure 5 jof-10-00281-f005:**
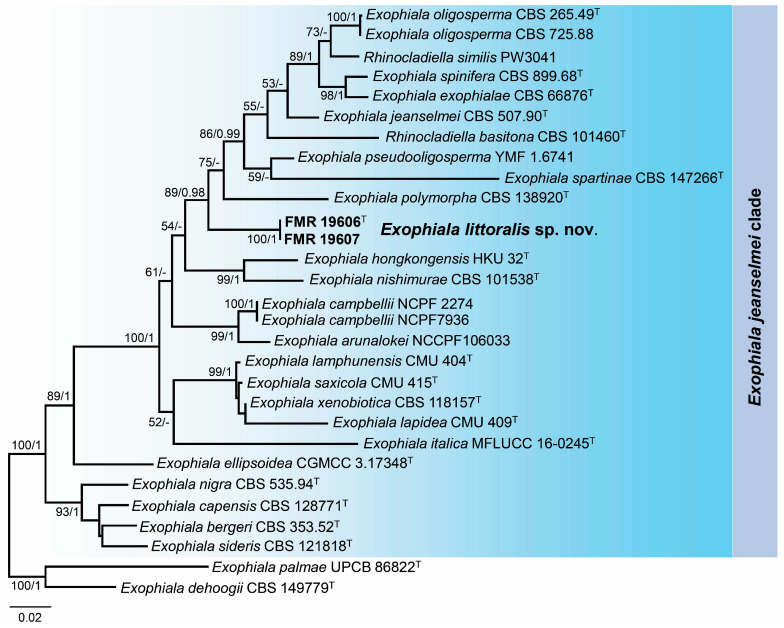
Phylogenetic tree inferred from a concatenated alignment of ITS and LSU sequences of 29 strains representing the *Exophiala jeanselmei*-clade. Numbers at the branches indicate support values (RAxML-BS/BI-PP) above 70%/0.95. The tree is rooted in *Exophiala dehoogii* CBS 149779 and *Exophiala palmae* UPCB 86822. The new species is highlighted in bold. ^T^ indicates ex-type strains. The scale bar represents the expected number of changes per site.

**Figure 6 jof-10-00281-f006:**
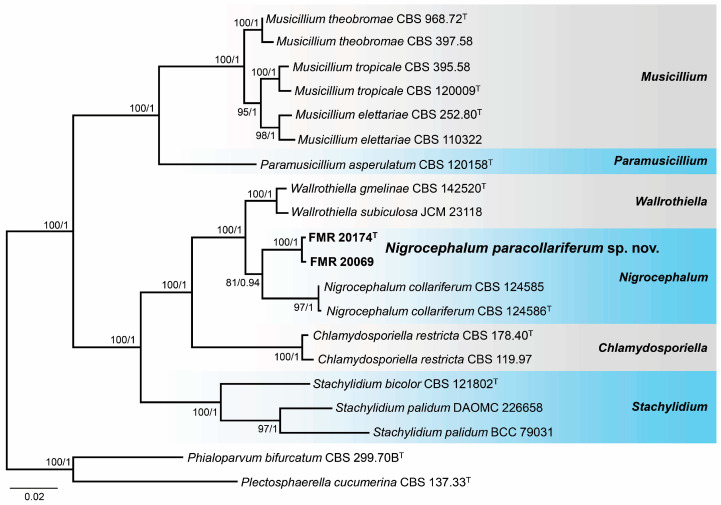
Phylogenetic tree was inferred from a concatenated alignment of ITS, LSU, *tef*1-α, and *rpb*2 sequences of 20 strains representing *Nigrocephalum* and related genera within *Plectosphaerellaceae*. Numbers at the branches indicate support values (RAxML-BS/BI-PP) above 70%/0.95. The tree is rooted in *Phialoparvum bifurcatum* CBS 299.70B and *Plectosphaerella cucumerina* CBS 137.33. The new species is highlighted in bold. ^T^ indicates ex-type strains. The scale bar represents the expected number of changes per site.

**Figure 7 jof-10-00281-f007:**
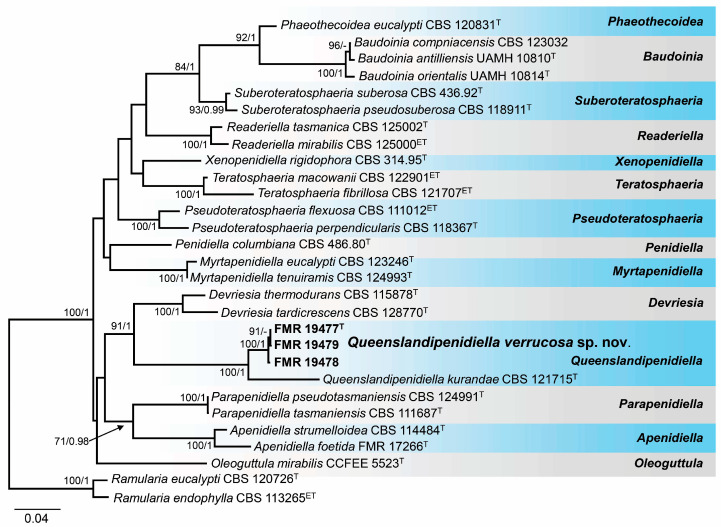
Phylogenetic tree was inferred from a concatenated alignment of ITS and LSU sequences of 27 strains representing *Queenslandipenidiella* and related genera within *Teratosphaeriaceae*. Numbers at the branches indicate support values (RAxML-BS/BI-PP) above 70%/0.95. The tree is rooted in *Ramularia eucalypti* CBS 120726 and *Ramularia endophylla* CBS 113265. The new species is highlighted in bold. ^T^ and ^ET^ indicate ex-type and ex-epitype strains, respectively. The scale bar represents the expected number of changes per site.

**Figure 8 jof-10-00281-f008:**
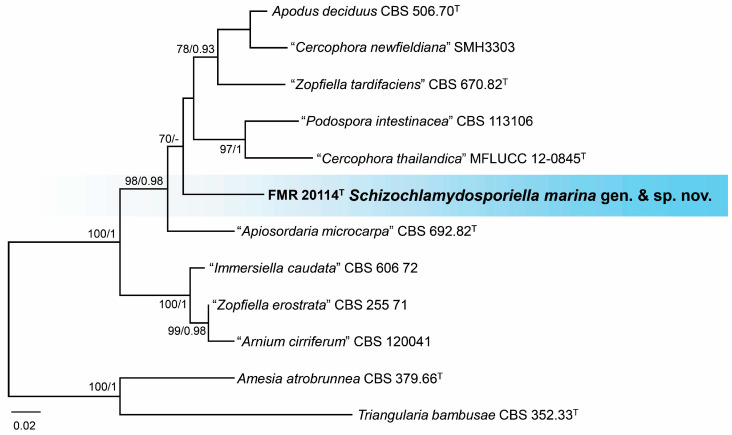
Phylogenetic tree was inferred from a concatenated alignment of ITS, LSU, and *rpb*2 sequences of 12 strains representing a clade in *Schizotheciaceae*. Numbers at the branches indicate support values (RAxML-BS/BI-PP) above 70%/0.95. The tree is rooted in *Amesia atrobrunnea* CBS 379.66 and *Triangularia bambusae* CBS 352.33. The new species is highlighted in bold. ^T^ indicates ex-type strains. Quote marks indicate strains with an unresolved taxonomy. The scale bar represents the expected number of changes per site.

**Figure 9 jof-10-00281-f009:**
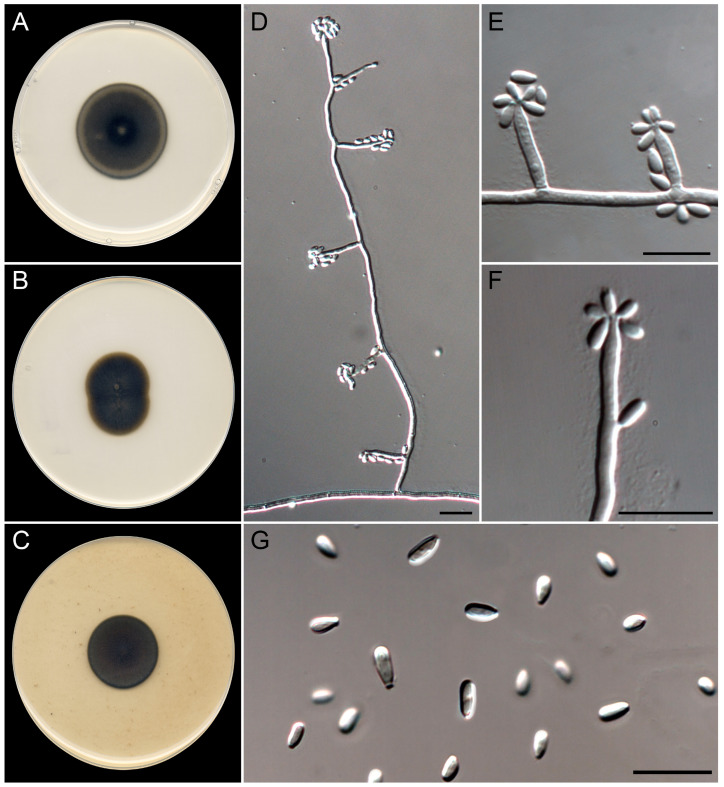
*Exophiala littoralis* (ex-type FMR 19606). (**A**–**C**) Colonies on PDA, MEA, and OA at 25 °C after 14 d. (**D**) Micronematous conidiophore. (**E**) Conidiogenous cells and conidia are borne directly from hyphae. (**F**). Detail of the apical annellated zone of the conidiogenous cells. (**G**) Conidia. Scale bars: 10 µm.

**Figure 10 jof-10-00281-f010:**
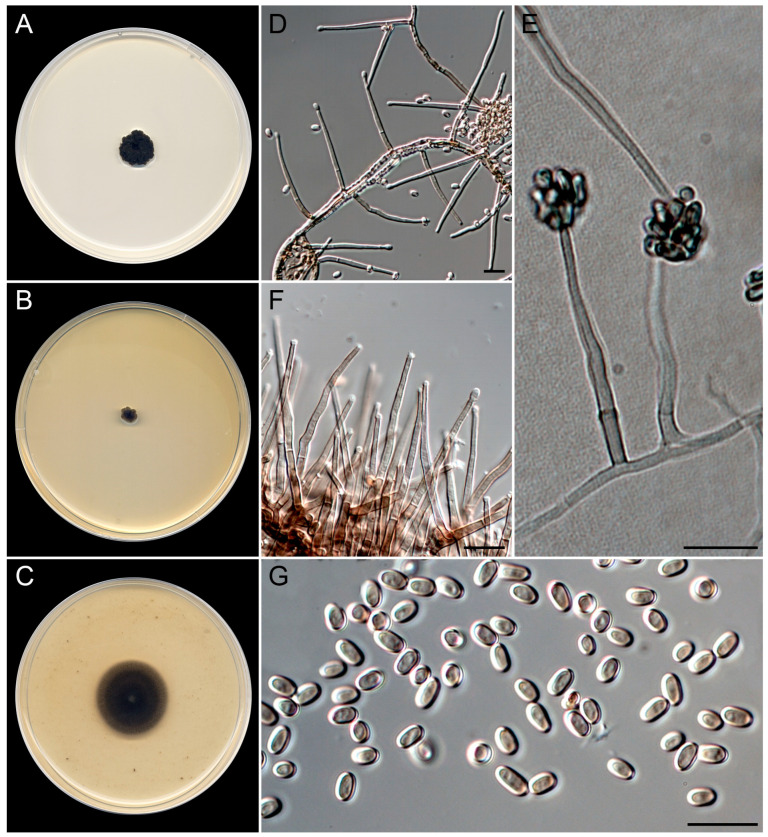
*Nigrocephalum paracollariferum* (ex-type FMR 20069). (**A**–**C**) Colonies on PDA, MEA, and OA at 25 °C after 14 d. (**D**) Micronematous conidiophores. (**E**) Conidiogenous cells with slimy heads of conidia. (**F**) Detail of the funnel-shaped collarettes in the apex of the conidiogenous cells. (**G**) Conidia. Scale bars: 10 µm.

**Figure 11 jof-10-00281-f011:**
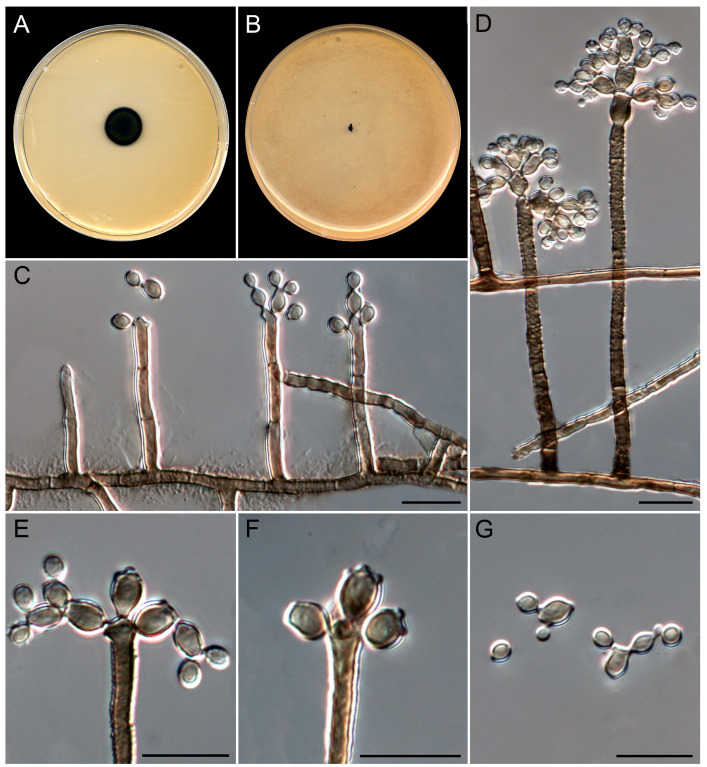
*Queenslandipenidiella verrucosa* (ex-type FMR 19477). (**A**,**B**) Colonies on MEA and OA at 25 °C after 14 d. (**C**) Development of the conidiophores. (**D**) Mature conidiophores: the conidiophore on the right shows an inflated conidiogenous cell. (**E**) Detail of the conidiogenous cell and two sets of ramoconidia. (**F**) Detail of the conidiogenous loci in ramoconidia. (**G**) Ramoconidia and conidia. Scale bars: 10 µm.

**Figure 12 jof-10-00281-f012:**
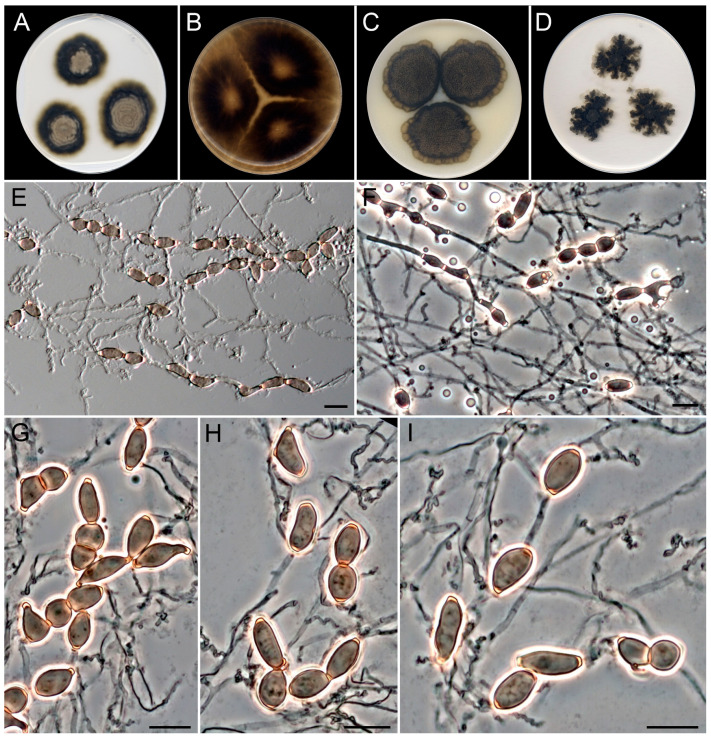
*Schizochlamydosporiella marina* (ex-type FMR 20114). (**A**–**D**) Colonies on MEA, OA, CMA, and PCA at 25 °C after 28 d. (**E**) Intercalary chains of chlamydospore-like cells. (**F**) Chains of chlamydospore-like cells under phase contrast microscopy (PCM). (**G**) Chains are composed of 0- and 1-septate chlamydospore-like cells (PCM). (**H**,**I**) Chlamydospore-like cells released by schizolythic secession show terminal, thick-walled scars (PCM). Scale bars: 10 µm.

**Table 1 jof-10-00281-t001:** Sets of primer combinations were used for DNA amplification and sequencing for each fungal family.

Locus	Primer Pairs	Taxa	Annealing Temp. (°C)	Orientation	Primer Reference
ITS–LSU (LSU D1-D3 region)	ITS5	Every strain	53	Forward	[36]
	LR5			Reverse	[37]
β-tubulin (*tub*2)	Bt2a	*Aspergillaceae*, *Bionectriaceae*, *Microascaceae*, *Trichocomaceae*	56	Forward	[38]
	Bt2b			Reverse	
	Btub2f	*Trichocomaceae*	56	Forward	[39]
	Btub4R			Reverse	
	T1	*Chaetomiaceae*, *Schizotheciaceae*	56	Forward	[40]
	Btub4R			Reverse	[39]
Traslation elongation factor-1α (*tef*1-α)	EF-983F	*Bionectriaceae*, *Plectosphaerellaceae*	57	Forward	[41]
	EF-2218R			Reverse	
	EF-728F	*Cladosporiaceae*, *Nectriaceae*	57	Forward	[42]
	EF-986R			Reverse	
RNA polymerase II second largest subunit (*rpb*2)	RPB2-5F	*Bionectriaceae*, *Chaetomiaceae*, *Cucurbitariaceae*, *Plectosphaerellaceae*, *Schizotheciaceae*, *Thyridariaceae*	60	Forward	[43]
	RPB2-7R			Reverse	
SSU	NS1	*Incertae sedis*	53	Forward	[36]
	NS4			Reverse	

**Table 2 jof-10-00281-t002:** Overview of the alignment parameters and phylogenetic models used for each genetic region in *Exophiala*, *Nigrocephalum*, *Queenslandipenidiella,* and *Schizochlamydosporiella*.

Dataset	Parameters	*Exophiala*	*Nigrocephalum*	*Queenslandipenidiella*	*Schizochlamydosporiella*
ITS	Lenth (bp)	625	511	529	545
	Conserved sites	382	374	284	376
	Variable sites	226	130	240	159
	Parsimony informative	147	93	204	75
	BI model	SYM+I+G	GTR+G	GTR+I+G	SYM+G
LSU	Lenth (bp)	873	791	751	827
	Conserved sites	801	714	568	730
	Variable sites	60	77	180	94
	Parsimony informative	38	42	139	45
	BI model	K80+I+G	K80+I	SYM+I+G	K80+I
*rpb*2	Lenth (bp)	-	743	-	840
	Conserved sites	-	484	-	460
	Variable sites	-	259	-	380
	Parsimony informative	-	219	-	223
	BI model	-	GTR+I+G	-	GTR+I+G
*tef*1-a	Lenth (bp)	-	787	-	-
	Conserved sites	-	624	-	-
	Variable sites	-	163	-	-
	Parsimony informative	-	132	-	-
	BI model	-	GTR+G	-	-
Concatenated	Lenth (bp)	1498	2832	1280	2212
	Conserved sites	1183	2196	852	1566
	Variable sites	286	629	420	633
	Parsimony informative	185	486	343	343

## Data Availability

Data are contained within the article and Appendix A.

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
