# Peer review of "Sunken Riches: Ascomycete Diversity in the Western Mediterranean Coast through Direct Plating and Flocculation, and Description of Four New Taxa"

_jof, 2024, doi:10.3390/jof10040281_

Round 1
Reviewer 1 Report
This study is well-designed and well-prepared. It is worthy of publishing but still needs some improvements. The language needs to be improved.
please refer to the MS with the comments

Author Response
Thank you for reviewing our study and offering feedback. We appreciate your suggestions to enhance the manuscript. We have thoroughly revised the language and incorporated all your corrections and suggentions indicated in the PDF original draft into the main text of the current (version R1). However, we think that lines 273-279 should be retained in text format to provide a characterization of each beach without the need to reference tables or figures. Figures 2 and 3 allow readers to access the genera and species recovered from each beach without having to consult the main text.

Reviewer 2 Report
Review Report for: Sunken Riches: Ascomycete Diversity in the Western Mediterranean Coast through Direct Plating and Flocculation, and Description of Four New Taxa.
Thanks to the editor and authors for allowing me to read this valuable paper. It is a well-written, very comprehensive manuscript that describes relevant work on the topic of mycological biodiversity. I believe that the manuscript has merit to be publishable after the authors review the indicated comments. My criticisms will be directed to the taxonomic identification and definition-delimitation of known and new species, since it is my area of expertise.
Major comments:
1. In the lines 201-203 the authors state: A maximum similarity level of >98% with ≥90% of sequence coverage with a type strain was used for species-level identification. In lines 208-209 the authors continues: Some of the strains that could not be identified at the species level with a BLAST search were subjected to phylogenetic analyses…
The identification of species using this type of BLAST phenetic criteria is a common conceptual error in microbiology. There is no such thing as “identifying at the species level with BLAST”, if that were possible there would no longer be a species problem. There is still no phylogenetic marker that solve ‘the problem of the species’ only using the similarity criteria and its pre-established cuts-off. Neither the complete SSU sequences with their more than 1600 bp, nor the ITS, LSU or any protein markers individually, have sufficient resolution to resolve species, not even sharing 100% identity. The proof of this is the counterexamples, it is possible to share 100% between two taxa in any of these barcodes, and still have room for speciation. To demonstrate whether the compared taxa are coalescent (same species) it is necessary to provide more evidence, for example, phylogenetic evidence that already adds another evolutionary corpus to the problem. So that, as a taxonomist I do not support this phenetic point as the sole criterion for define consespecific contexts and I state that phylogenetics is not optional (as the authors show, applying it only in cases that are not resolved with blast; blast barely allows us to select phylogenetic neighbors...). The first thing I recommend to the authors is to confirm from phylogenetics each of the species-level assignments they made in Table S1, mainly those that were assigned with a single marker.
2. The second serious point in the methodology to define new species is the total absence of a conceptual corpus in the manuscript for it. The authors do not adhere to any conceptual approach that allows testing the species hypothesis. A methodical taxonomic exercise involves a hypothesis driven study where several sub-hypotheses that theoretically support the processes of Speciation or Coalescence are rejected or accepted; That makes the taxonomic exercise a rigorous process. As recommended by Aime et al. How to publish a new fungal species, or name, version 3.0. IMA Fungus 12, 11 (2021): authors should provide a statement of the guiding species concept used to delimit newly proposed species…
What are the biological hypotheses that the authors adhere to define new species? It is evident that they use phenotypic and phylogenetic criteria, but the authors fail to offer an integrative vision towards the problem of the species. As a guide the authors can see a modern exercise evaluating the species hypothesis in fungi at: Ide-Pérez (2024). Exophiala chapopotensis sp. nov., an extremotolerant black yeast from an oil-polluted soil in Mexico; phylophenetic approach to species hypothesis in the Herpotrichiellaceae family. Plos one, 19(2), e0297232.
3. The third point is: This proposal lacks evolutionary-molecular vision in its approach to new species; which would be an added weight to the species hypothesis and taxonomists should use more frequently in our proposals. The multi-locus phylogenies in Figures 5-8 leave little to the imagination, these are all good results. But phylogenetic is still a hypothesis. Running evolutionary-molecular tests of speciation on the phylogenies would be a second piece of evidence that would add support to their proposal. In summary, I recommend the authors perform the following molecular speciation tests to confirm all their proposals for new species:
Test1- Generalized mixed Yule-coalescent (GMYC)
Test2-Poisson tree processes (bPTP)
Test3- Assemble Species by Automatic Partitioning (ASAP)
I suggest modify the lines 201-203 on the statements: A maximum similarity level of >98% with ≥90% of sequence coverage with a type strain was used for species-level identification. and lines 208-209: Some of the strains that could not be identified at the species level with a BLAST search were subjected to phylogenetic analyses…
In order to be conceptually precise and not measleading
Author Response
“Thanks to the editor and authors for allowing me to read….. the manuscript has merit to be publishable after the authors review the indicated comments. My criticisms will be directed to the taxonomic identification and definition-delimitation of known and new species, since it is my area of expertise.”
Response: Thank you for reviewing our study and offering valuable feedback. We have carefully revised the methodology and results sections, as well as the language, following your suggestions. Below, we offer detailed responses to each comment provided.
Major comments:
- “In the lines 201-203 the authors state: A maximum similarity level of >98% with ≥90% of sequence coverage with a type strain was used for species-level identification. In lines 208-209 the authors continues: Some of the strains that could not be identified at the species level with a BLAST search were subjected to phylogenetic analyses… The identification of species using this type of BLAST phenetic criteria is a common conceptual error in microbiology. There is no such thing as “identifying at the species level with BLAST”, if that were possible there would no longer be a species problem. There is still no phylogenetic marker that solve ‘the problem of the species’ only using the similarity criteria and its pre-established cuts-off. Neither the complete SSU sequences with their more than 1600 bp, nor the ITS, LSU or any protein markers individually, have sufficient resolution to resolve species, not even sharing 100% identity. The proof of this is the counterexamples, it is possible to share 100% between two taxa in any of these barcodes, and still have room for speciation. To demonstrate whether the compared taxa are coalescent (same species) it is necessary to provide more evidence, for example, phylogenetic evidence that already adds another evolutionary corpus to the problem. So that, as a taxonomist I do not support this phenetic point as the sole criterion for define consespecific contexts and I state that phylogenetics is not optional (as the authors show, applying it only in cases that are not resolved with blast; blast barely allows us to select phylogenetic neighbors...). The first thing I recommend to the authors is to confirm from phylogenetics each of the species-level assignments they made in Table S1, mainly those that were assigned with a single marker.”
Response: In response to the reviewer's comment, we acknowledge the limitations associated with using BLAST phenetic criteria for species identification in microbiology. We agree that the description provided in lines 201-209 of the main text may have been misleading, and as such, we have revised and improved the methodology section accordingly. It's important to clarify that the strains listed in Table S1 were not solely delineated based on a BLAST search. Initially, each strain underwent morphological characterization, followed by delineation using barcoding sequences. The BLAST search was employed to determine related accessions for our strains, with a focus on type sequences of accepted taxa. We conducted BLAST searches using diagnostic barcodes recommended in the most recent phylogenies for each taxon, even for strains assigned using a single marker. While we acknowledge the value of phylogenetic analysis in preventing misinterpretation of molecular data, the large number of strains recovered in our study necessitated a time-effective approach to diversity sorting. Furthermore, coalescent analyses are increasingly utilized to synonymize names within species-rich groups like Aspergillus, Penicillium, and Talaromyces, where in recent years several species had been described despite the high similarity between protein coding regions. Therefore, we think that the methodology outlined in the text provides a confident assignment of our strains to the species mentioned in Table S1.
- “The second serious point in the methodology to define new species is the total absence of a conceptual corpus in the manuscript for it. The authors do not adhere to any conceptual approach that allows testing the species hypothesis. A methodical taxonomic exercise involves a hypothesis driven study where several sub-hypotheses that theoretically support the processes of Speciation or Coalescence are rejected or accepted; That makes the taxonomic exercise a rigorous process. As recommended by Aime et al. How to publish a new fungal species, or name, version 3.0. IMA Fungus 12, 11 (2021): authors should provide a statement of the guiding species concept used to delimit newly proposed species… What are the biological hypotheses that the authors adhere to define new species? It is evident that they use phenotypic and phylogenetic criteria, but the authors fail to offer an integrative vision towards the problem of the species. As a guide the authors can see a modern exercise evaluating the species hypothesis in fungi at: Ide-Pérez (2024). Exophiala chapopotensis nov., an extremotolerant black yeast from an oil-polluted soil in Mexico; phylophenetic approach to species hypothesis in the Herpotrichiellaceae family. Plos one, 19(2), e0297232.”
Response: Following your suggestion, we have updated the text to refer to the consolidated species concept, which integrates the morphological, ecological, and phylogenetic species concepts. (Quaedvlieg, W., et al. "Introducing the consolidated species concept to resolve species in the Teratosphaeriaceae." Persoonia-Molecular Phylogeny and Evolution of Fungi 33.1 (2014): 1-40.).
- “The third point is: This proposal lacks evolutionary-molecular vision in its approach to new species; which would be an added weight to the species hypothesis and taxonomists should use more frequently in our proposals. The multi-locus phylogenies in Figures 5-8 leave little to the imagination, these are all good results. But phylogenetic is still a hypothesis. Running evolutionary-molecular tests of speciation on the phylogenies would be a second piece of evidence that would add support to their proposal. In summary, I recommend the authors perform the following molecular speciation tests to confirm all their proposals for new species: Test1- Generalized mixed Yule-coalescent (GMYC), Test2-Poisson tree processes (bPTP), Test3- Assemble Species by Automatic Partitioning (ASAP)”
Response: We appreciate the reviewer's suggestions for additional analyses of our data. Following the recommendation, we conducted bPTP, ASAP, and GMYC analyses, and we have provided the results in a separate folder (Speciation analyses). Overall, these analyses support our hypotheses regarding novel species. Specifically, our putative novel species hypotheses were supported by the bPTP analysis, with all MCMC runs converging and the posterior probabilities supporting our strains as independent lineages. The ASAP analysis, which relies on single locus alignments, generally supported our species hypotheses. However, regions like ITS or RPB2 provided stronger support compared to more conserved regions like LSU, which tended to cluster strains together. Regarding the GMYC analysis, none of our runs were statistically significant. Despite this, each of our hypotheses was determined as an independent lineage, except for Schizochlamydosporiella (Schizotheciaceae). We attribute this result to the limited number of genetic markers available for strains accepted in Schizotheciaceae, highlighting the limitations of speciation tests.
These tests are commonly used to assess populations but rely on the use of several strains per species hypothesis, with each hypothesis represented by genetically different strains. However, due to limited genetic information and the small number of curated strains available for many species, particularly in genera with few accepted species, the performance of these tests can be hindered. Therefore, while bPTP, ASAP, and GMYC support our novel species hypotheses, we think their inclusion in the manuscript is not warranted given the limitations of our data. However, we are open to feedback from the reviewer and editor on these results. We are willing to consider adding these analyses as supplementary material if deemed appropriate.
Detail comments
“I suggest modify the lines 201-203 on the statements: A maximum similarity level of >98% with ≥90% of sequence coverage with a type strain was used for species-level identification. and lines 208-209: Some of the strains that could not be identified at the species level with a BLAST search were subjected to phylogenetic analyses…In order to be conceptually precise and not measleading”
Response: These lines were modified in accordance to the reviewer’s previous suggestions.

Round 2
Reviewer 2 Report
No further comments
No further comments